# "It becomes more difficult when people don't empathize with us": COVID-19-related stigmatization experienced by survivors in Nepal

Buna Bhandari[1,2,3]*, Poshan Thapa[4], Amit Timilsina[5], Rajiv Ranjan Karn[6], Haider Ali[7], Ashley Hagaman[8], Archana Shrestha[9]

1 Central Department of Public Health, Tribhuvan University Institute of Medicine, Kathmandu, Nepal, 2 Dwyer School of Health Sciences, Indiana University South Bend, Indiana, United States of America, 3 Department of Global Health and Population, Harvard T.H Chan School of Public Health, Boston, Massachusetts, United States of America, 4 School of Population and Global Health, McGill University, Quebec, Canada, 5 Research and Community Development Centre, IPAS, Kathmandu, Nepal, 6 Department of Research, Biratnagar Eye Hospital, Biratnagar, Nepal, 7 Health Division, Biratnagar Metropolitan, Biratnagar, Nepal, 8 Social and Behavioural Sciences Department, Yale School of Public Health, New Haven, Connecticut, United States of America, 9 Community Health Department, Kathmandu University Dhulikhel Hospital, Dhulikhel, Nepal

☙ These authors contributed equally to this work.
* buna.bhandari@gmail.com

## Abstract

The COVID-19 pandemic caused widespread social disruption, with stigma emerging as a significant challenge for individuals who survived infection. This qualitative study explored the forms, drivers, and impacts of COVID-19-related stigma among survivors in Eastern Nepal. In-depth interviews were conducted with 15 COVID-19 survivors who had reported stigma in a preceding cross-sectional survey. Due to pandemic-related restrictions, interviews were conducted over the phone. Data were analysed thematically following the process outlined by Braun and Clarke. COVID-19 stigma was multifaceted, including social rejection, internalized stigma, and discriminatory practices by community members. Key drivers of stigma included self-directed fear of infection and death, misinformation and limited awareness about COVID-19 transmission and prevention, and a fragile health system and policy responses. Although COVID-19-related stigma may have declined as the pandemic evolved, the findings illustrate how stigma can emerge rapidly during health emergencies that can have social consequences related to trust, disclosure, and help-seeking behavior in future crisis. The study highlights the importance of outbreak preparedness strategies that integrate clear communication, strengthened health system capacity, and social protection measures to mitigate stigma and its harms during future public health crises.

**Data availability statement:** All relevant data are within the paper and its Supporting information files.

**Funding:** The study was funded by Nepal Health Research Council provincial grant [Grant number 778]. The funders had no role in study design, data collection and analysis, and preparation of this manuscript.

**Competing interests:** The authors have declared that no competing interests exist.

## Introduction

The Coronavirus disease (COVID-19) pandemic profoundly impacted individuals' psychological and mental well-being worldwide. During the pandemic, factors such as anxiety, fear, insecurity, social and emotional isolation, financial hardship, and inadequate health infrastructure compounded the challenges faced by affected populations. Among these challenges, stigma emerged as a critical issue for COVID-19 survivors, exacerbating their physical, emotional, and social vulnerabilities [1,2]. Stigma experienced by survivors was widely reported across countries, with prevalence estimates as high as 79% in China [3], 46% in Ghana [4], and 43.3% in Japan [5]. A pooled analysis of stigma related to infectious diseases, Severe Acute Respiratory Syndrome (SARS), Influenza A virus subtype H1N1, Middle East respiratory syndrome coronavirus (MERS), Zika virus, Ebola virus, and COVID-19, reported an estimated prevalence of 34%, highlighting the widespread nature of stigma during infectious disease outbreaks [6].

Historically, epidemics and pandemics have consistently resulted in severe social consequences, most notably stigma and discrimination [7]. This pattern has been observed in several infectious diseases, including Human Immunodeficiency Virus (HIV), tuberculosis (TB), SARS, Ebola, and H1N1 influenza, with COVID-19 being no exception [8]. Stigma has a detrimental impact on individuals, as demonstrated by its association with HIV. Evidence from HIV research shows that stigma contributes to depression, anxiety, and emotional distress, which in turn leads to diminished quality of life and reduced adherence to treatment protocols [9]. Understanding stigma in the context of COVID-19 among survivors is therefore crucial, given its far-reaching short and long-term consequences.

The World Health Organization (WHO) defines social stigma as the "negative association between a person or group of people who share certain characteristics and a specific disease." Stigma manifests through labelling, stereotyping, discrimination, differential treatment, and loss of social status [10]. These manifestations can have severe consequences, including job termination, social exclusion, abandonment, and physical violence [10]. Furthermore, stigma drives individuals to conceal their illness, delays healthcare-seeking behaviour, and discourages the adoption of preventive measures [10,11]. These repercussions extend beyond individuals, affecting families, communities, and the healthcare system. In Nepal, research on stigma related to TB, HIV, leprosy, and mental health has found significant impacts on patients' health, access to care, and quality of life [11].

Understanding the stigma faced by COVID-19 survivors was particularly important during the pandemic period, given documented experiences in Nepal and other settings [12,13]. The nature and magnitude of stigma are highly context-specific, affecting individuals, families, communities, and societal structures at multiple levels [14]. A comprehensive understanding of stigma among COVID-19 survivors can help identify its underlying drivers, develop targeted interventions, and help promote support systems. While several studies in Nepal have examined stigma directed towards healthcare providers and migrant returnees during the pandemic [15], fewer have focused on COVID-19 survivors. Therefore, the primary objective of this study was to explore the forms, impact and drivers of stigma experienced by COVID-19 survivors in the context of Nepal.

## Materials and methods

### Study design and population

This qualitative study was conducted between May 19 and July 24, 2021, as part of a larger mixed-method research project in Biratnagar Metropolitan City, Koshi Province, Nepal. The first phase of the study began with a cross-sectional survey of 385 COVID-19 survivors who were registered in the government reporting system. This first study aimed to estimate the prevalence of mental health problems experienced by survivors (unpublished). Findings from this quantitative survey indicated that 24.7% (n = 101) of respondents reported experiencing some form of stigma.

For this qualitative study, survivors who reported experiencing any form of stigma in the previous quantitative survey and had provided informed consent for further participation were considered eligible for inclusion in the sampling frame. All 101 survivors had provided their consent to be contacted for participation in future research. Twenty participants were selected purposively by two researchers (BB and PT), ensuring variation in factors such as gender, age, education, occupation and average family size, to facilitate a deeper exploration of their experiences and the underlying factors contributing to stigma faced by them. During follow-up, five declined to participate, citing personal reasons such as lack of time and lack of private space for a phone interview, resulting in 15 participants being interviewed. No specific sample size was predetermined, as recruitment was constrained by feasibility challenges in approaching and enrolling participants during the heightened COVID-19 pandemic. The study reporting was guided by the COREQ checklist to enhance transparency and completeness [16].

### Data collection tool and method

To explore the stigma experiences of COVID-19 survivors, an in-depth interview guide was developed, drawing on the socio-ecological framework. The socio-ecological framework, which guided this study, was appropriate because it captures how stigma operates and interacts across individual, interpersonal, and structural levels, enabling a comprehensive examination of the multifaceted experiences of COVID-19 survivors in Nepal [17]. The interview topic guide was initially developed in Nepali in discussion with all co-authors, then translated to English. The overall process was guided by the qualitative methods expert co-author (AH), who has extensive experience working on similar topics across diverse populations globally, including Nepal. The interview guide was designed to examine the nature of stigma, drivers (including underlying structural, social, and individual-level factors), and its impact on COVID-19 survivors. It was pre-tested with two participants, and these interviews were not included in the final analysis.

Given the strict lockdown during the second wave of COVID-19 in Nepal, and to ensure the safety and convenience of both participants and RAs, all interviews were conducted over the phone. We did not use video platforms such as Zoom because many participants lacked access to the internet or computers or the necessary skills to use them. A total of 15 in-depth interviews were conducted with participants who reported experiencing any form of stigma. To assure the quality of the data collected, research Assistants (RAs) with prior qualitative research experience and familiarity with the local context conducted the interviews. This also helped ensure neutrality, thus limiting the risk of researchers' bias. Before the data collection process, the RAs were thoroughly trained on the study objectives, interview tools, and protocols to ensure quality in data collection. The RAs first contacted the participants who were purposively selected. All chosen participants were accessible via phone, either on their own device or on a family member's device. During the initial contact, the study objectives, methods, and expectations were explained. As the interviews were conducted virtually, no compensation was provided to participants, and this was clearly communicated during the first contact. If participants consented to take part, they were given two options: to participate in the interview on the same day or to choose an alternative date and time based on their convenience. All interviews were conducted in Nepali. After consent was obtained, the interviews were recorded, and each lasted an average of 35–45 minutes.

## Data analysis

The audio-recorded in-depth interviews were transcribed verbatim in Nepali and subsequently translated into English by the RAs for data analysis. Translation validation was ensured by performing a back-translation of two transcripts. To ensure confidentiality, pseudonyms were assigned to participants in the transcripts.

Thematic analysis was employed to analyse the study data, following the six-step process outlined by Braun and Clarke [18]. The analysis process involved multiple stages, including familiarising with the data, generating initial codes using an inductive coding approach, searching for themes, reviewing, defining, naming themes, and producing the final report. To enhance the rigour and reliability of the analysis, three researchers (BB, PT and AT) independently read and coded the transcripts. Inductive coding was conducted to generate data, and codes were described and grouped into sub-themes based on shared characteristics. The sub-themes were then aggregated to form overarching themes, as presented in Table 2. The researchers regularly discussed the codes, themes, and interpretations to compare and refine them. NVivo version 12 software [19] was used to organize and manage the qualitative data.

## Ethical consideration

The study was approved by the institutional ethics committee of the Nepal Health Research Council (Reg. No. 2897). As the interviews were conducted remotely, informed verbal consent was obtained from all participants, an approach which was approved by the ethics committee. Before each interview, the RAs re-read the full consent script to participants and allowed time for any questions or clarifications. Once participants confirmed their understanding, verbal consent was audio-recorded. To maintain confidentiality, the consent recordings were stored separately from the interview recordings. RAs also confirmed that the participants were in a comfortable and private space to share their experiences, particularly given that many were confined at home with family members during the lockdown periods. All interview recordings and transcripts were stored securely, and only de-identified data were shared among the authors. To ensure anonymity, the names of individuals, places, and institutions were replaced with pseudonyms in the transcripts and subsequent analysis.

## Trustworthiness

We used Lincoln and Guba's (1989) four criteria: credibility, dependability, confirmability, and transferability to establish trustworthiness in this study [20]. To ensure credibility, we pilot-tested the interview guide and conducted in-depth interviews in Nepali to collect rich information from participants. Research assistants, trained and familiar with the local context, conducted interviews, applying probing and clarification techniques to strengthen data quality. Rich verbatim quotations are presented in the results to illustrate the findings. To enhance dependability, we provide a transparent description of the sampling, recruitment, data collection, and analysis, enabling others to assess the consistency of the process. To increase confirmability, coding was carried out independently by three researchers (BB, PT, and AT), with discrepancies resolved through discussion. The researchers also shared codes with the RAs involved in data collection throughout the coding process. For transferability, we provided thick descriptions of the study participants and the research process, allowing readers to assess the applicability of our findings to other settings with similar contexts.

## Results

### Characteristics of study participants

Table 1 presents the demographic characteristics of study participants who reported experiencing some form of stigma and discrimination. Nearly half of the participants (46.6%) were aged 31–45, and 53% were male. Most participants (66.6%) had an education level above higher secondary level, and 73.3% were employed at the time of the interview. The majority of the participants (93%) reported having fewer than six family members, as shown in Table 1.

**PLOS One**

**Table 1. Characteristics of study participants included in the in-depth interviews (n = 15).**

| S. N | Age | Gender | Education | Occupation | Family size |
|------|-----|--------|-----------|------------|-------------|
| IDI1 | 21 | Male | High School | Computer operator | 4 |
| IDI2 | 35 | Male | Masters | Part-time teacher | 3 |
| IDI3 | 50 | Male | Masters | Marketing | 4 |
| IDI4 | 27 | Female | Bachelors | Student | 3 |
| IDI5 | 39 | Male | Secondary level education | Student | 3 |
| IDI6 | 55 | Male | High School | Teacher | 5 |
| IDI7 | 40 | Female | Primary education | Unemployed | 4 |
| IDI8 | 38 | Female | High School | Business | 3 |
| IDI9 | 58 | Male | Bachelor | Accountant | 11 |
| IDI10 | 38 | Female | High School | Business | 3 |
| IDI11 | 46 | Female | Bachelor | Teacher | 3 |
| IDI12 | 26 | Male | Masters | Officer in a Ministry | 6 |
| IDI13 | 38 | Female | Primary education | Ward assistant | 3 |
| IDI14 | 27 | Female | Master Completed | NGO worker | 4 |
| IDI15 | 38 | Male | Master's degree in medicine | Officer | 6 |

### Themes and sub-themes from the data

The qualitative analysis generated 53 codes, which were organized into seven sub-themes and three overarching themes. These themes describe how stigma was experienced and enacted among COVID-19 survivors during the COVID-19 period. Table 2 summarises the themes, sub-themes, and corresponding codes derived from the data.

1. Dimensions of multifaceted stigma

This theme captures the various dimensions of stigma experienced by COVID-19 survivors in Nepal, focusing on their personal encounters with stigma and broader societal practices.

1.1. Stigma experience among COVID-19 survivors and their families

All participants in this study reported experiencing stigma directed at themselves and, in some cases, their families from community members and neighbours following disclosure of their COVID-19 status. Participants described stigma manifesting in various forms and intensities, such as social avoidance, verbal labelling and altered interactions during their infection. One participant drew parallels between their experience and stigma historically associated with conditions, such as HIV and TB:

*There is a shopkeeper nearby, who used to say (neighbour who used to take care of them) "don't take the money", or "don't go near them".This behaviour made me feel like those persons discriminated due to AIDS or TB, which used to happen in Nepal decades ago. It did make me feel really bad.* (IDI 2)

Community-level stigma was widespread once participants' COVID-19 status was revealed. Participants described community members blocking access to homes, evicting tenants, and labelling homes and streets as *Corona Ghar* or *Corona bato* ("Home infected by Coronavirus" or "Street infected by Coronavirus"). While some friends maintained contact remotely due to the lockdown situation, others avoided interaction entirely.

*I felt very discriminated... when others found out about my COVID-19 status, they stopped contacting me. When I informed them that I had tested positive, they stopped visiting my home.* (IDI 8)

**Table 2. Themes and sub-themes derived from the data.**

| Theme | Sub-themes | Codes |
|---|---|---|
| Dimensions of multi-faceted stigma | Stigma experiences among COVID-19 survivors and their families | Internalized stigma due to self-directed fear of infection, perceived stigma, strained social relationships, reduced social support, intersecting stigma, discrimination in community and workplace settings, and stigma directed towards family members of COVID-19 survivors. |
| | Stigma practices associated with COVID-19 | Forceful quarantine, being treated as untouchable, receiving humiliating taunts, unnecessary attention, hesitance to communicate, stereotyping of COVID-19 infection and infected individuals, media-driven, and bias in healthcare seeking. |
| Impact of COVID-19-related stigma | Well-being of persons with COVID-19 infection | Limited access to medicine, lack of support for survivors, anxiety, fear of death, perceived physical weakness, mental stress, exhaustion, lack of confidence, and loneliness. |
| | Fragile interpersonal trust and support | Lack of social protection, absence of caregivers during isolation, and inability to support pregnant and postpartum women. |
| Drivers of COVID-19-related stigma | Fear of Covid-19 | Fear of contracting the virus, fear of death, inaccessible treatment, social media-amplified fear, high death rates, fear of recurrence, and anticipated discrimination due to/during illness. |
| | Limited information about COVID-19 | Lack of knowledge, misconceptions, misinformation on social media, misunderstandings about disease transmission, lack of trust in COVID-19 tests, and limited awareness of protocols and vaccines. |
| | Fragile health system and policy responses | Unclear COVID-19 protocols, limited healthcare capacity, lack of equipment, insufficient healthcare personnel, inadequate public health programs, lack of government preparedness, neglect of the social dimension in response, and delayed screening and treatment. |

Workplace-related stigma was commonly reported among employed participants. Survivors described being blamed for potential disease transmission and experiencing altered interactions from colleagues, even post-recovery and negative test results. A female participant shared:

*Conflicts arose among the staff in our office, and they blamed each other. I had been practising self-isolation and refrained from interacting with others. Since I isolated myself early on, no one could accuse me of spreading the coronavirus to them. But they would swiftly close the door when I was outside.* (IDI 3)

Participants reported stigma directed towards their family members by the community. Members of the community even directed negative comments and discriminatory behaviour towards children whose parents contracted COVID-19:

*Children also faced it. When they went to buy Cetamol (medicine), people said, 'their mother is infected, why did you take that money, etc.' The child thought, 'Our mom is infected, so this money will not work,' and they became sad. They behaved discriminately against my sons.* (IDI 7)

Direct discrimination from immediate family members was largely absent. However, one participant described internalizing stigma within the family context, where self-directed fear of infection shaped his behaviour toward a critically ill family member. Although family members enacted no discrimination, internalized fear results in self-imposed distancing during illness.

*Because the situation was so bad at home, everybody was scared. I was scared, too; being a son, I did not meet my mother even though she was so critical after the COVID-19 infection.* (IDI 9)

Several participants also reported a lack of support from government officials, local leaders and community members. Despite seeking assistance, some individuals were denied help and, in some cases, evicted by landlords while facing financial difficulties. One participant stated:

*Some individuals who were in home isolation after testing positive for COVID-19 sought assistance from key figures in society, including the police, but received no help. Instead, their landlords compelled them to leave their homes and had to stay at a hotel, even though they were facing financial difficulties.* (ID10)

1.2. Stigma Practice associated with COVID-19

Participants described multiple enacted stigma *(experience of discrimination, or negative treatment due to perceived identity, or condition)*. These include being treated as untouchables, receiving humiliating remarks and being singled out by neighbours. Stigma was perceived as more pronounced in urban areas than in rural regions. One participant shared:

*I didn't get any help at the place where I reside. My friends living far away used to call, talk, and help when needed. But in the village where I live, there was no social support.* (IDI 10)

A recurring experience involved shopkeepers refusing to accept cash directly from COVID-19 survivors or discouraging others from receiving their money. One participant recounted the discrimination her children faced.

*My sons went to buy medicine when I became severely ill. When they went to buy medicine, they said their mother was Corona positive and refused to take the money. When my sons walked in the road, they made remarks saying their mother was Corona-positive.* (IDI 7)

Another reported stigma practice involved publicly disclosing individuals' COVID-19 status, often resulting in forced quarantine in hotels, even for individuals facing financial hardships. Participants also noted that discrimination often persisted after recovery, with continued reluctance from others to engage with them. Health workers, including doctors and nurses, faced similar experiences and could not quarantine themselves in their homes. A male participant shares:

*People even refused to allow infected health workers working in hospitals as doctors or nurses to stay in their homes. There was a widespread fear that they would transmit the virus.* (IDI 12)

2. Impact of COVID-19-related stigma

This theme describes the consequences of stigma on survivors' physical, psychological, and social well-being.

2.1. Well-being of the person with COVID-19 infections

Many participants reported that their physical and mental health deteriorated due to the constant discrimination and stigma they faced. Survivors described physical weakness from the infection, compounded by inadequate care, stress, mental exhaustion, and heightened anxiety driven by fears of reinfection and death.

*[..] and at that time we had heard a lot of news saying many people had lost their lives abroad because of COVID, and I felt like I would not be able to live now. Corona will take my life too. I had so much stress at that time, which was beyond my thinking.* (IDI 13)

Additionally, friends and community members' lack of empathetic behaviour further undermined participants' well-being. This resulted in feelings of loneliness, abandonment and a diminished sense of confidence:

*Within our family, some of us were infected while others were not infected with COVID-19. The community people suspected (falsely) that the other non- infected members were infected too. The community members used to avoid the non-infected family members while walking through our house and talk behind our backs. They (the non-infected family members) felt that discrimination. No one cared to come and ask how we are dealing with the situation.* (IDI 10)

### 2.2. Fragile interpersonal trust and support

The lockdown imposed during the COVID-19 pandemic disrupted family connections, leaving many participants isolated from their loved ones. Migrant individuals and those away from their families expressed that the absence of family support hindered their recovery. They believed that being with family could have provided the care and assistance necessary for physical and mental well-being.

*Maybe it's because I didn't get any support then. I was far from my family during that COVID-19 situation; I think that if my family had been there, they could have helped me get medicine whenever I needed it.* (IDI 14)

Discrimination and stigma also led to severe consequences, including the death of individuals due to a lack of community support and resources. One participant recalled a neighbour who passed away because he could not receive even the most basic care:

*There was a lot of discrimination. Near my home, due to a lack of care, he was even unable to boil water by himself. At the time of taking him to a hospital, he died.* (IDI-12)

## 3. Drivers of COVID-19-related stigma

### 3.1. Fear of COVID-19

The fear of contracting the virus and dying from it was one of the most commonly cited reasons for stigma. Participants highlighted how the lack of guaranteed treatment or vaccines during the early phase of the pandemic contributed to fear and avoidance of COVID-19 survivors:

*Fear!! People had limited knowledge about the disease, its transmission, and its causes. They were afraid of the COVID-19 outbreak, and there was a lack of understanding about disease prevention.* (IDI 10)

Participants also described the fear of being forcibly quarantined by authorities, which led many to hide their COVID-positive status, disclosing it only to close family members:

*During that time, the main problem was that the people were scared of being taken by the police and used to hide the fact that they had the disease. If any case were found, the area would have been sealed, people would have hesitated to go near the patients, and even the whole family would have faced the same.* (IDI 10)

Social media played a significant role in amplifying stigma, often sensationalizing and misrepresenting isolation practices and sharing unscientific information, which negatively impacted survivors:

*At the time of being diseased, people are mentally weak and left alone. Social media posts about people in isolation affected patients mentally. It was unpractical and unscientific.* (IDI 14)

## 3.2. Limited Information regarding COVID-19

Participants attributed stigma to widespread misinformation and limited awareness about COVID-19 transmission, testing, and preventive measures. Many individuals held misconceptions, such as believing that COVID-19 survivors would inevitably die or that the virus remained in the body despite a negative test result:

*Corona is transmitted by not wearing a mask properly, staying close, or shaking hands. People should understand these things to stop discriminating against COVID-19 patients.* (IDI 6)

Lack of accurate information contributed to discriminatory behaviours, and internalized fear, with some participants describing feelings of shame and hesitancy in social interactions:

*Discrimination occurs due to a lack of awareness, nothing more than that. Instead of discrimination, safety measures should be in place, such as maintaining distance. It's okay not to hug or be too close. Initially, I felt shy and fearful when facing others, fearing negative comments. Even after receiving a negative test report, I still had a lingering fear.* (IDI 12).

Several participants emphasized that inadequate dissemination of clear protocols and guidelines contributed to confusion and stigma, and they reflected the importance of community education and awareness during health emergencies.

*Had society been well-informed and educated, such discrimination wouldn't have occurred. I know ummm, for a disease like corona (COVID-19), it is necessary to keep some distance, but there is a standard protocol of 3 meters or 6 meters. All of that happened because of a lack of proper information.* (IDI 2)

## 3.3. Fragile health system and policy responses

Participants frequently cited weakness in the health system as a major driver of stigma. Limited hospital beds, a shortage of equipment, and untrained healthcare workers (for such a pandemic) compromised COVID-19 care and heightened fear within the community, which fueled stigma:

*Even patients were not getting beds in the hospital after spending a long time in the emergency department. I don't think there was any support from the community health system or government.* (IDI 14)

*The fear of infection and the lack of guaranteed treatment at a health facility made the discrimination worse.* (IDI 2)

Participants also highlighted the government's lack of preparedness in responding to the pandemic. Issues included inadequate enforcement of COVID-19 protocols, ineffective communication of guidelines, and a lack of concrete plans to address myths and misconceptions. One participant recounted:

*We know everything and have a lockdown for prevention, but it still lacks safety measures. Talking about my experience, I returned home from Kathmandu the last time before this second lockdown (second wave). At that time, it was said that vehicles were running under safety measures, maintaining distance, and sanitising, but it was not like that. There were as many passengers as before COVID-19, and without maintaining social distancing. It was so packed, and bus fare was also double or triple the regular amount. And that's not how to do it; it was not satisfying.* (IDI 4)

A widespread lack of trust in the government's response was evident among participants. One participant shared:

*The government did nothing; everything is just empty promises and lectures. It's just written on paper. Everything is based on books and paper, so we should not expect the government to do this or that or say that the government didn't do this or that. We ourselves should be aware.* (IDI 3)

## Recommendations for future preparedness

Participants reflected on their experiences during the COVID-19 pandemic and offered recommendations to strengthen preparedness for future public health emergencies. They emphasized the need for the government to prioritize advanced planning and coordinated response that explicitly addresses stigma alongside biomedical containment strategies. Participants highlighted the importance of clear policies, strategies, and procedures to prevent discrimination and protect the rights of affected individuals during a health crisis.

One participant stressed the need for a formal mechanism to address stigma and discrimination during outbreaks.

*Policy about social discrimination should be prepared. It should provide justice for the affected people in a time of need.* (IDI 15)

Participants also suggested that the government improve its preparedness by learning from the COVID-19 experience to strengthen social protection measures during future pandemics. Suggestions included timely financial assistance, access to nutrition support, and the provision of uninterrupted access to essential medicines to reduce fear, vulnerability, and stigma among affected populations.

*The government should look after coronavirus patients and survivors from low-income backgrounds and provide financial support and supplementation. Some financially unstable coronavirus patients and survivors have to fight against the disease with just a normal diet (meal of dal bhat (rice and lentils)), so a little help there in a proper supplement package and nutrition or through direct financial support would be better.* (IDI 5)

Several participants reflected on the role of health insurance during the pandemic, noting that it provided some, albeit partial, financial relief during periods of illness and isolation. Although insurance coverage did not fully offset the broader economic consequences associated with COVID-19, participants viewed it as an important form of financial protection that helped manage healthcare-related expenses. These experiences highlighted the value of strengthening and expanding health insurance coverage and broader social protection mechanisms to prepare for future pandemics.

*One thing I had done was insurance. I got the insurance money, and it did cover some expenses at that time. But it is difficult to manage things now.* (IDI 10)

Participants highlighted the importance of timely and clear communication during health emergencies. They emphasized that awareness and information campaigns are essential to counter misinformation, reduce fear, and prevent stigma during outbreaks. In this context, participants specifically pointed to vaccination, noting that while vaccines were available during the pandemic, not all individuals were adequately reached or informed.

*Awareness campaigns should run so swiftly that everyone should understand this disease. It could also be about vaccination. I have gotten my vaccination, but many haven't; even for that, awareness is a necessity.* (IDI 1)

## Discussion

This qualitative study provides valuable insights into the multifaceted experiences of stigmatization among COVID-19 survivors in Nepal. It explores various forms of stigma, impacts, drivers, and potential response considerations. These findings should be interpreted within the specific temporal and social context of the COVID-19 pandemic, particularly periods characterized by uncertainty regarding transmission, lockdown restrictions, and evolving public health guidance. The stigma described by participants reflects experiences during the acute phase of the outbreak and may have diminished as population-level knowledge, vaccination coverage and health system response have improved. Therefore, the relevance of these findings lies in the post-pandemic context in understanding the rapid emergence of stigma during emergencies and identifying response elements that may prevent similar harms during future outbreaks.

The study findings emphasize enacted and associated stigma faced by COVID-19 survivors, particularly interpersonal discrimination from neighbours and community members. Similar patterns of stigma and discrimination toward COVID-19 patients have been documented in other countries [21–23]. For instance, studies from India reported high levels of enacted and internalized stigma among COVID-19 survivors [22]. Research from China also highlighted significant stigma experienced by COVID-19 patients, mainly manifesting as social rejection, financial insecurity, internalized shame, and social isolation, closely aligning with the experiences reported in this study [23]. While some participants interpreted their experiences through comparisons with stigma historically associated with conditions such as HIV or TB, COVID-19-related stigma in this context appears more closely linked to crisis conditions and is likely to follow a more time-limited trajectory [24].

Our study participants reported various forms of enacted stigma, including being treated as untouchables, enduring humiliating remarks, and having their residential areas labelled as "Corona home" or "Corona Street." Families of COVID-19 survivors also experienced social rejection and discriminatory behaviours from community members. Similar patterns of labelling, discrimination against family members, and social rejection have been documented in studies conducted in Uganda and Jordan [25,26]. Some studies have reported more severe manifestations of stigma, such as physical violence and verbal abuse [22,27,28]. However, participants in our study did not report such extreme experiences.

This study found no reports of direct discriminatory behaviour from immediate family members, whereas most participants experienced stigma from neighbours and community members, consistent with the findings of Regmi et al. [29]. Except for one participant who described internalized stigma driven by self-directed fear of infection, family members were portrayed as supportive rather than discriminatory. Community-level stigma had substantial consequences for participants' physical and mental well-being, consistent with findings from other settings [30].

The reasons behind stigma and discrimination identified in this study were multifaceted and context specific. Fear of infection, recurrence, and death emerged as primary drivers of stigma, alongside misinformation, inadequate awareness, and limited knowledge about COVID-19 transmission, management, and prevention. The role of misinformation, mainly through digital and social media platforms, has been well-documented during the pandemic [31–33], and appears to have amplified fear-driven response and discriminatory behaviour, especially in low- and middle-income countries (LMICs).

One of the key findings of this study is the link between a fragile health system and policy gaps with the stigmatization of COVID-19 survivors. Participants highlighted several systemic shortcomings, including inadequate preparedness, the lack of comprehensive plans and policies, limited healthcare capacity, insufficient social and behaviour change communication, and the absence of explicit strategies to address the social consequences of the pandemic. These findings align with evidence from other settings, which shows that weak health systems, inadequate governance, and poor communication contributed to stigma during the pandemic [24,27,34,35].

Although COVID-19-related stigma has likely declined in the post-pandemic period, experiences of discrimination, forced isolation, and lack of support may have longer-term implications for trust in institutions and health-seeking behaviour. Such experiences could influence willingness to disclose symptoms or engage with health services during

future outbreaks or for other communicable diseases, particularly in contexts characterized by fear and misinformation. These potential implications require further investigation.

## Strengths and limitations

A key strength of this study is its exploration of the multi-dimensional nature of COVID-19 stigmatization. This approach allowed for a comprehensive exploration of stigma across individual, interpersonal, and structural levels, as well as its drivers and impacts. The study's findings provide valuable guidance for policymakers, government officials, and community stakeholders seeking to integrate socio-ecological approaches into strategies to address stigma during public health emergencies such as COVID-19. In addition, the inclusion of participants from diverse backgrounds facilitated a richer understanding of how survivors across various social groups experienced stigma.

The study also has a few limitations. First, data collection was conducted through virtual interviews, which may have limited the ability to observe and capture non-verbal cues in participants' responses. Second, the self-reported nature of the data introduces potential for recall and social desirability biases, as participants may not fully or accurately recount their experiences of stigma. However, to address this limitation, the study employed a rigorous methodological approach, including pretesting of the interview tool, comprehensive training of research assistants, and probing techniques during interviews to elicit detailed responses. Finally, the findings are specific to the context of Biratnagar Metropolitan City in Nepal and may not be directly transferable to other regions or populations.

## Conclusion

This study documents the forms and drivers of stigma experienced by COVID-19 survivors in Nepal during the acute phase of the pandemic. Survivors reported social rejection, internalized stigma, discrimination practices by community members, and negative labelling, shaped by fear of infection, a fragile health system and gaps in policy responses. These experiences adversely affected their physical and mental well-being. Although COVID-19-related stigma has likely diminished in the post-pandemic context, the findings remain important for understanding how stigma can emerge rapidly during health emergencies that can have social consequences related to trust, disclosure, and help-seeking behavior in future crisis. The findings emphasize the need for outbreak preparedness strategies that integrate clear and timely communication, strengthened health system capacity, and social protection mechanisms that reduce vulnerability and fear. By situating COVID-19-related stigma within this temporal context while identifying transferable lessons, this study contributes evidence to inform more socially responsive and equitable responses to future pandemics and emerging infectious disease threats.

## Supporting information

**S1 File. Additional information.**
(DOC)

## Acknowledgments

We would like to acknowledge the Health Section of the Biratnagar Metropolitan City Office team for providing data, all the Research Assistants who helped with data collection, transcription, and translation, and the Nepal Health Research Council for funding this study.

## Author contributions

**Conceptualization:** Buna Bhandari, Poshan Thapa, Amit Timilsina, Rajiv Ranjan Karn, Haider Ali, Ashley Hagaman, Archana Shrestha.

**Data curation:** Rajiv Ranjan Karn, Haider Ali, Archana Shrestha.

**Formal analysis:** Buna Bhandari, Poshan Thapa, Amit Timilsina, Rajiv Ranjan Karn, Haider Ali, Ashley Hagaman, Archana Shrestha.

**Funding acquisition:** Buna Bhandari, Poshan Thapa, Rajiv Ranjan Karn, Haider Ali, Ashley Hagaman, Archana Shrestha.

**Investigation:** Buna Bhandari, Poshan Thapa, Amit Timilsina, Rajiv Ranjan Karn, Haider Ali, Ashley Hagaman, Archana Shrestha.

**Methodology:** Buna Bhandari, Poshan Thapa, Amit Timilsina, Rajiv Ranjan Karn, Haider Ali, Ashley Hagaman, Archana Shrestha.

**Project administration:** Buna Bhandari, Poshan Thapa, Amit Timilsina, Rajiv Ranjan Karn, Haider Ali, Archana Shrestha.

**Resources:** Buna Bhandari, Poshan Thapa, Amit Timilsina, Rajiv Ranjan Karn, Archana Shrestha.

**Software:** Buna Bhandari, Poshan Thapa, Archana Shrestha.

**Supervision:** Buna Bhandari, Poshan Thapa, Rajiv Ranjan Karn, Ashley Hagaman, Archana Shrestha.

**Validation:** Buna Bhandari, Poshan Thapa, Ashley Hagaman, Archana Shrestha.

**Visualization:** Buna Bhandari, Poshan Thapa, Rajiv Ranjan Karn, Ashley Hagaman, Archana Shrestha.

**Writing – original draft:** Buna Bhandari, Poshan Thapa, Amit Timilsina, Rajiv Ranjan Karn, Haider Ali, Ashley Hagaman, Archana Shrestha.

**Writing – review & editing:** Buna Bhandari, Poshan Thapa, Amit Timilsina, Rajiv Ranjan Karn, Haider Ali, Ashley Hagaman, Archana Shrestha.

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
