## [Decision Letter · Decision Letter 0]

12 Aug 2025

Dear Dr.  Bhandari,

Thank you for submitting your manuscript to PLOS ONE. After careful consideration, we feel that it has merit but does not fully meet PLOS ONE’s publication criteria as it currently stands. Therefore, we invite you to submit a revised version of the manuscript that addresses the points raised during the review process.

We look forward to receiving your revised manuscript.

Kind regards,

Surangi Jayakody, MBBS, MSc, MD

Academic Editor

PLOS ONE

Journal Requirements:

2. In the ethics statement in the Methods, you have specified that verbal consent was obtained. Please provide additional details regarding how this consent was documented and witnessed, and state whether this was approved by the IRB

3. We note that your Data Availability Statement is currently as follows:

“All relevant data are within the manuscript and its Supporting Information files.”

Additional Editor Comments:

**Please address the comments of Reviewer 1 as stated in the attached manuscript and the comments of Reviewer 2 given below.**

Reviewer's Responses to Questions

**Comments to the Author**

1. Is the manuscript technically sound, and do the data support the conclusions?

Reviewer #1: Yes

Reviewer #2: Yes

2. Has the statistical analysis been performed appropriately and rigorously?

Reviewer #1: N/A

Reviewer #2: N/A

3. Have the authors made all data underlying the findings in their manuscript fully available?

Reviewer #1: Yes

Reviewer #2: No

4. Is the manuscript presented in an intelligible fashion and written in standard English?

Reviewer #1: Yes

Reviewer #2: Yes

Reviewer #1: I have added comments to the paper. Please check the Word document. Good luck! The major comment from me is to clearly state and explain what stigma framework you used, discuss the broader implications of the findings, as some stigma experiences might not be relevant now and even non-existent.

Reviewer #2: In this research article, authors explored the multidimensional stigma related to COVID 19 and its ecological impact on survivors in Eastern Nepal.

As a qualitative study, authors used a appropriate methodological approach and written it in scientific manner.

However,

The title seems to be longer. If it is concise it would be more attractive.

There are some methodological aspects to be revisited.

1.This is a subsequent study component conducted following a survey according to the authors. When choosing 15 out of 101 eligible participants (even though with purposive sampling method), it is clear if inclusion and exclusion criteria are mentioned if there are.

2. It is great if there is a justification for the sample size of 15. Whether this is according to the saturation or predetermined sample size …??

3. Study instruments –

According to the given facts, authors used a semi structured interviewer guide. The rigour of the facts of the study will depend on the validity of the study instrument used for data collection. Hence, to be more transparent, better to mention

Who developed it?

What scientific methods occupy in the preparation of it?

Did validity check? If so, how?

4. Data collection-

Authors mentioned that the data collected via telephone calls. It is recommended to describe how actually the interviews took place. The way of taking consent, time of the interview (approximate), what happened in the interview, Was it recorded...?

Additionally, did all the chosen participants have access to mobile phones?

If so, did all the selected participants respond?

Were there any payments to the participants for participation in this interview.

5. Ethical considerations-

Other than the informed consent, what other ethical aspects were concerned with?

**Do you want your identity to be public for this peer review?** For information about this choice, including consent withdrawal, please see our Privacy Policy

Reviewer #1: No

Reviewer #2: No

---

## [Author Response · Author response to Decision Letter 1]

18 Sep 2025

We are grateful to the editors and reviewers for their constructive feedback, which has significantly strengthened our manuscript. In response, we have clarified our methods, and thoroughly revised the manuscript, where requested. A detailed, point-by-point response to each comment is included below and attached in a separate file as well.

Comment from editor:

In the ethics statement in the Methods, you have specified that verbal consent was obtained. Please provide additional details regarding how this consent was documented and witnessed, and state whether this was approved by the IRB

Response: Thank you for your advice. We have added the following information to the ethical consideration section. (See page 6, line 137-148)

The study was approved by the Institutional Ethics Committee of the Nepal Health Research Council (Reg. No. 2897). As the interviews were conducted remotely, informed verbal consent was obtained from all participants, an approach which was approved by the ethics committee. Before each interview, the RAs re-read the full consent script to participants and allowed time for any questions or clarifications. Once participants confirmed their understanding, verbal consent was audio-recorded. To maintain confidentiality, the consent recordings were stored separately from the interview recordings. RAs also checked if participants were in a comfortable space to share their perspectives and experiences, particularly given that many were confined at home with family members during the lockdown. All interview recordings and transcripts were stored securely, and only de-identified data were shared among the authors. To ensure anonymity, real names of people, places, and institutions were replaced with pseudonyms in the transcripts and subsequent analysis.

Data Availability

The provided data available statement stands true.

Comment from reviewer:

Reviewer #1: I have added comments to the paper. Please check the Word document. Good luck! The major comment from me is to clearly state and explain what stigma framework you used, discuss the broader implications of the findings, as some stigma experiences might not be relevant now and even non-existent.

Response: Thank you for your constructive feedback. We have clarified and explained about the stigma framework and boarder implications of the findings in the revised manuscript as presented detailed below

1. Appropriate references

Response: This comment has been addressed.

2. As stigma is understood, described and theorized in different ways it is important that you mention what theory/framework was used for your analysis. For an example you mention 'enacted stigma' in the results section, but it is not defined anywhere. You mention drivers but have not defined what 'drivers' mean?

Response: Thank you for highlighting this. We have added more details on the theory/framework and clarified key terminology in the revised manuscript (lines 92–101 and 235–236)

To explore the stigma experiences of COVID-19 survivors, a semi-structured in-depth interview guide was developed, drawing on the socio-ecological framework. The socio-ecological framework, which guided this study, was appropriate as it captures how stigma operates and interacts across individual, interpersonal, and structural levels, reflecting the analysis of the multi-faceted experiences of COVID-19 survivors in Nepal [17].

The tool was designed to examine nature, drivers (underlying structural, social, and individual-level factors), and the impact of stigma on COVID-19 survivors informed by the Socioecological framework to guide the domains/questions and analysis.

3. Data quality is important in qualitative research. Please do explain how you maintained data quality and trustworthiness and the validity and reliability of findings?

Response: We have explained about the data quality and trustworthiness and the validity and reliability of findings as follows in the revise manuscript. (Page 7, line 150-163)

We used Lincoln and Guba’s (1989) four criteria: credibility, dependability, confirmability, and transferability to establish trustworthiness in this study. To ensure credibility, we pilot-tested the semi-structured interview guide and conducted in-depth interviews in Nepali to collect rich information from participants. Research assistants, trained and familiar with the local context, conducted interviews, applying probing and clarification techniques to strengthen data quality. Rich verbatim quotations are presented in the results to illustrate the findings. To enhance dependability, we provide a transparent description of the sampling, recruitment, data collection, and analysis, enabling others to assess the consistency of the process. To increase confirmability, coding was carried out independently by three researchers (BB, PT, and AT), with discrepancies resolved through discussion. The researchers also shared codes with the research assistants (RAs) involved in data collection throughout the coding process. For transferability, we provided thick descriptions of the study participants (Table 1) and the research process, allowing readers to assess the applicability of our findings to other settings with similar contexts.

4. Authors mention that the sampling frame was people who have had experienced stigma and provided consent. More details are needed how you selected the 15 from the above sampling frame. What criteria was used for purposive sampling?

Response: Thank you. We have provided more details as follows in the revised manuscript. (Page 4, line 81-93)

For this qualitative study, survivors who reported experiencing stigma in the quantitative survey and had provided informed consent for further participation were considered eligible for inclusion in the sampling frame. All 101 survivors had provided their consent to be contacted for participation in future research. The participants were selected purposively by two researchers (BB and PT), maximizing variation in factors such as gender, age, education, occupation, and average family size, to facilitate a deeper exploration of their lived experiences and the underlying factors contributing to stigma among COVID-19 survivors in Nepal. During follow-up, five declined to participate, citing personal reasons such as lack of time and lack of private space to take part in a phone interview, resulting in 15 participants being interviewed. No specific sample size was predetermined, as recruitment was constrained by feasibility challenges in approaching and enrolling participants during the heightened period of the COVID-19 pandemic.

5. You might want to tweak this sentence to introduce the socio-ecological framework and summaries it in one-two sentences.

Response: Thank you for your feedback this has been well addressed in the line 92-96 as follows.

To explore the stigma experiences of COVID-19 survivors, a semi-structured in-depth interview guide was developed, drawing on the socio-ecological framework. The socio-ecological framework, which guided this study, was appropriate as it captures how stigma operates and interacts across individual, interpersonal, and structural levels, reflecting the analysis of the multi-faceted experiences of COVID-19 survivors in Nepal [16].

6. It would be best to add a additional file with characteristic of each 15 participants separately rather than as a summary table. For your quotes to be meaningful there should be a reference for a reader to go back and understand their background (age, gender etc).

Response: This comment has been addressed and table with details characteristics of each participants has been provided in the revised manuscript ( see table 1).

7. I wonder whether you should change the wording of this sub theme. Since some of it does not 'exists' anymore such as forced quarantine? Simply stigma practices associated with covid-19 could work better

Response: Thank you for this feedback. We have made the necessary changes accordingly in the revised result section.

8. As the data is old and while findings on structural stigma can be important to highlight system issues, findings on enacted stigma etc might be old and not relevant anymore. authors must critique more on how to use such data in the future and what is the real implication of having this data out now? Especially considering broader health related stigma.

Response: Thank you for this important feedback. We underscore the implications of our data and findings through an illustration of structural stigma and its impact on health system has been explained in lines 366-383 later in the paragraph.

9. In the results section you declare that 'All participants in this study reported experiencing stigma from their family and community members, manifesting in various forms and intensities during their infection" isn't this contradictory?

Response: While all participants in this study reported experiencing stigma, the source of stigma can be different. While few participants experienced stigma from their family, majority of the participants experience stigma from neighbors and friends. This has been explained in the line 269-276

10. Explain why you didn't use virtual techniques such as zoom or other video conferencing platforms as they could have given you a chance to observe non-verbal cues.

Response: We have clarified our approach in detail in the revised manuscript as follows ( see page 5, line 103-106)

Considering the strict lockdown during the second wave of COVID-19 in Nepal, and to ensure the safety and convenience of both participants and RAs, all interviews were conducted over the phone. We did not use video platforms such as Zoom, as many participants did not have access to the internet, computers or the necessary skills to use such platforms.

Reviewer #2:

In this research article, authors explored the multidimensional stigma related to COVID 19 and its ecological impact on survivors in Eastern Nepal. As a qualitative study, authors used a appropriate methodological approach and written it in scientific manner. However, The title seems to be longer. If it is concise, it would be more attractive. There are some methodological aspects to be revisited.

Response: Thank you for the constructive feedback. We have concise the title and addressed each comment as follows.

1) This is a subsequent study component conducted following a survey according to the authors. When choosing 15 out of 101 eligible participants (even though with purposive sampling method), it is clear if inclusion and exclusion criteria are mentioned if there are. It is great if there is a justification for the sample size of 15. Whether this is according to the saturation or predetermined sample size …??

Response: Thank you for your feedback. The criteria for selection, the purpose of purposive sampling has been explained in the line 78-89.

For this qualitative study, survivors who reported experiencing stigma in the quantitative survey and had provided informed consent for further participation were considered eligible for inclusion in the sampling frame. All 101 survivors had provided their consent to be contacted for participation in future research. The participants were selected purposively by two researchers (BB and PT), maximizing variation in factors such as gender, age, education, occupation, and average family size, to facilitate a deeper exploration of their lived experiences and the underlying factors contributing to stigma among COVID-19 survivors in Nepal. During follow-up, five declined to participate, citing personal reasons such as lack of time and lack of private space to take part in a phone interview, resulting in 15 participants being interviewed. No specific sample size was predetermined, as recruitment was constrained by feasibility challenges in approaching and enrolling participants during the heightened period of the COVID-19 pandemic.

2) Study instruments –

According to the given facts, authors used a semi structured interviewer guide. The rigor of the facts of the study will depend on the validity of the study instrument used for data collection. Hence, to be more transparent, better to mention Who developed it? What scientific methods occupy in the preparation of it? Did validity check? If so, how?

Response: We have clarified all of the approach of development and pretesting of the tool in the revised manuscript as follows (Line 92-103)

The interview topic guide was developed originally in Nepali in discussion with all co-authors then translated to English and was guided by the qualitative methods expert co-author (AH), who has extensive experience working on similar topics across diverse populations globally, including Nepal. The tool was designed to examine nature, drivers, and the impact of stigma on COVID-19 survivors informed by the Socioecological framework to guide the domains/questions. The semi-structured Nepali interview guide was pre-tested among two participants, and these interviews were not included in the final analysis.

3) Data collection: Authors mentioned that the data collected via telephone calls. It is recommended to describe how actually the interviews took place. The way of taking consent, time of the interview (approximate), what happened in the interview, was it recorded...? Additionally, did all the chosen participants have access to mobile phones?

If so, did all the selected participants respond? Were there any payments to the participants for participation in this interview.

Response: Thank you for your feedback this has been explained in the methodology section line 104-121

Considering the strict lockdown during the second wave of COVID-19 in Nepal, and to ensure the safety and convenience of both participants and RAs, all interviews were conducted over the phone. We did not use video platforms such as Zoom, as many participants did not have access to the internet, computers or the necessary skills to use such platforms. A total of 15 in-depth interviews were conducted with participants who reported experiencing some form of stigma. To assure the quality of the data collected, research Assistants (RAs) with prior qualitative research experience and familiarity with the local context conducted the interviews. This also helped to ensure neutrality thus limiting the chance of researcher's bias. Before the data collection process, the RAs were thoroughly trained on the study objectives, interview tools, and protocols to ensure quality in data collection. The RAs first contacted the participants who were purposively selected. All chosen participants were accessible via phone, either through their own device or that of a family member. During the initial contact, the study objectives, methods, and expectations were explained. As the interviews were conducted virtually, no compensation was provided to participants, and this was clearly communicated during the first contact. If participants consented to take part, they were given two options: to participate in the interview on the same day or to choose an alternative date and time based on their convenience. All interviews were conducted in Nepali. After consent, the interviews were recorded, and each lasted an average of 35 to 45 minutes.

5. Ethical considerations- Other than the informed consent, what other ethical aspects were concerned with?

Response: We have added more details in the revised manuscript as follows in line 138-149

The study was approved by the Institutional Ethics Committee of the Nepal Health Research Council (Reg. No. 2897). As the interviews were conducted remotely, informed verbal consent was obtained from all participants, an approach which was approved by the ethics committee. Before each interview, the RAs re-read the full consent script to participants and allowed time for any questions or clarifications. Once participants confirmed their understanding, verbal consent was audio-recorded. To maintain confidentiality, the consent recordings were stored separately from the interview recordings. RAs also checked if participants were in a comfortable space to share their perspectives and experiences, particularly given that many were confined at home with family members during the lockdown. All interview recordings and tran

---

## [Decision Letter · Decision Letter 1]

10 Nov 2025

Dear Dr. Bhandari,

We look forward to receiving your revised manuscript.

Kind regards,

Surangi Jayakody, MBBS, MSc, MD

Academic Editor

PLOS ONE

Journal Requirements:

Reviewers' comments:

Reviewer's Responses to Questions

**Comments to the Author**

Reviewer #1: (No Response)

Reviewer #2: All comments have been addressed

2. Is the manuscript technically sound, and do the data support the conclusions?

Reviewer #1: No

Reviewer #2: Yes

3. Has the statistical analysis been performed appropriately and rigorously?

Reviewer #1: N/A

Reviewer #2: Yes

4. Have the authors made all data underlying the findings in their manuscript fully available?

Reviewer #1: Yes

Reviewer #2: Yes

5. Is the manuscript presented in an intelligible fashion and written in standard English?

Reviewer #1: Yes

Reviewer #2: Yes

Reviewer #1: In one place, you say ‘All participants in this study reported experiencing stigma from their families and community members, manifesting in various forms and intensities during their infection’ and then “No participants reported

direct discrimination from immediate family members”. This is confusing and contradictory. If people have not faced stigma from family, then do not mention it as such in the first sentence.

The code you have ‘discrimination by family and friends, ’ also problematic, as you move on to say there was no discrimination. If this is the case, then term in as a ‘Lack of discrimination by family and friends’. Please go through the codes and reword as appropriate.

I do not think the authors have understood my previous comment about the need to discuss the findings of this study in relation to the current COVID-19-free context? Authors should address the context-specific nature of COVID-19 stigma, as it may now be almost non-existent. unlike the stigma surrounding, for example, HIV/AIDS or NTDs. This is because COVID-19-related stigma is likely to be temporary and does not exist now? Another angle could be how or whether these stigma experiences of these patients will affect help-seeking behaviours for other diseases as well? Will they have long-term socio-cultural effects? These need to be critiqued.

The conclusion of this study needs to be thoroughly revised.

In the Conclusion, the authors say “these findings emphasize the urgent need for comprehensive interventions to combat stigma and discrimination” but now that COVID is over, is the stigma still there? Is it still relevant to suggest something like that? What you need to suggest is how these findings can be used for future preparedness of such outbreaks.

Again, I question the relevance of this sentence to the current context. The key recommendations include implementing mass education and awareness campaigns to improve knowledge about disease transmission, prevention, and control, establishing platforms for accessing accurate

information, strengthening reporting systems, enhancing the capacity of health institutions, and developing robust social protection mechanisms to support individuals facing stigma.’ Are people still not aware of COVID? Is this suggestion relevant to the current context?

Reviewer #2: All the comments of the previous review have been addressed adequately by the authors in this submission.

The manuscript of the qualitative study on "COVID-19 stigmatisation and its impact on survivors in Nepal" is written in scientifically and addressed the multifaceted stigma in a coherent way.

**Do you want your identity to be public for this peer review?** For information about this choice, including consent withdrawal, please see our Privacy Policy

Reviewer #1: No

Reviewer #2: No

---

## [Author Response · Author response to Decision Letter 2]

17 Dec 2025

Response to Reviewer’s comments – PONE-D-24-58430R1

We thank the editor and reviewers for their time, careful assessment, and constructive feedback which have significantly improve the quality of our manuscript. All comments have been addressed thoroughly, and point-by-point responses to all the comments are provided below

1. Reviewer #1: In one place, you say ‘All participants in this study reported experiencing stigma from their families and community members, manifesting in various forms and intensities during their infection’ and then “No participants reported

direct discrimination from immediate family members”. This is confusing and contradictory. If people have not faced stigma from family, then do not mention it as such in the first sentence.

Response: Thank you for highlighting this inconsistency. We agree that the original wording was unclear and potentially misleading. We have revised the manuscript to clearly distinguish between stigma experienced from community members and experiences within the family context.

In the Results section, we now state that all participants reported experiencing stigma directed at themselves and, in some cases, their families from community members and neighbours following disclosure of their COVID-19 status. We further clarify that direct discrimination from immediate family members was largely absent, except one participant who described internalized stigma within the family context, where self-directed fear of infection shaped his behaviour rather than overt discriminatory actions. These clarifications are reflected consistently in both the Results and Discussion sections of the revised manuscript. (Line 216-223,429-435)

2. The code you have, ‘discrimination by family and friends, ’ is also problematic, as you move on to say there was no discrimination. If this is the case, then term in as a ‘Lack of discrimination by family and friends’. Please go through the codes and reword as appropriate.

Response: Following this review, we refined the wording of several codes to better reflect the underlying data. Importantly, we removed any code that implied direct discrimination by immediate family members, as the data indicated that family-level stigma was limited to internalized stigma in one participant rather than overt discriminatory behaviour. As such, a separate code for family discrimination was not retained.

Under Theme 1, the revised codes now include: internalized stigma due self-directed fear of infection, perceived stigma, strained social relationships, reduced social support, intersecting stigma, discrimination in community and workplace settings, and stigma directed towards family members of COVID-19 survivors. These refinements are presented in Table 2. The revisions reflect improved clarity and precision in coding and do not alter the study results or their interpretation.

We appreciate the reviewer’s thoughtful guidance, which has strengthened the conceptual clarity of the analysis.

3. I do not think the authors have understood my previous comment about the need to discuss the findings of this study in relation to the current COVID-19-free context? Authors should address the context-specific nature of COVID-19 stigma, as it may now be almost non-existent. unlike the stigma surrounding, for example, HIV/AIDS or NTDs. This is because COVID-19-related stigma is likely to be temporary and does not exist now? Another angle could be how or whether these stigma experiences of these patients will affect help-seeking behaviors for other diseases as well? Will they have long-term socio-cultural effects? These need to be critiqued.

Response: Thank you for this clarification. We agree that COVID-19-related stigma is context-specific and likely to be time-limited, and this is now explicitly addressed in the revised discussion section.

The discussion situates the findings within the early and acute phases of the COVID-19 pandemic, shaped by uncertainty, self-directed fear of infection, and restrictive public health measures. It acknowledges that such stigma may no longer be prominent in the current context. We also distinguish COVID-19-related stigma from more enduring forms of stigma, such as those associated with HIV/AIDS or neglected tropical diseases. Rather than implying persistence of COVID-19 stigma, the revised discussion frames the relevance of the findings in terms of lessons for future public health emergencies, including implications for risk communication, continuity of essential health services, and potential effects on help-seeking behaviours during future outbreaks. [Line 451-456, 401-409,479-483-511]

4. The conclusion of this study needs to be thoroughly revised.

In the Conclusion, the authors say “these findings emphasize the urgent need for comprehensive interventions to combat stigma and discrimination” but now that COVID is over, is the stigma still there? Is it still relevant to suggest something like that? What you need to suggest is how these findings can be used for future preparedness of such outbreaks. Again, I question the relevance of this sentence to the current context. The key recommendations include implementing mass education and awareness campaigns to improve knowledge about disease transmission, prevention, and control, establishing platforms for accessing accurate

information, strengthening reporting systems, enhancing the capacity of health institutions, and developing robust social protection mechanisms to support individuals facing stigma.’ Are people still not aware of COVID? Is this suggestion relevant to the current context?

Response: Thank you for highlighting this important issue. We agree that framing COVID-19-related stigma as ongoing would not be appropriate in the current context. In response, we have substantially revised the conclusion to remove language implying an urgent need to address ongoing COVID-19-related stigma.

The revised conclusion now explicitly acknowledges that COVID-19-related stigma has likely diminished in the post-pandemic period. Recommendations are no longer framed as current COVID-19 interventions but instead emphasize how the findings can inform future outbreak preparedness and response, including transferable lessons related to communication, health system capacity, and social protection during public health emergencies. (Line 476-489)

In response to the reviewer’s emphasis on the post-pandemic context, we also reviewed the manuscript more broadly. We revised the abstract and introduction sections to ensure that the study is clearly framed as examining stigma during the acute phase of COVID-19, with relevance for future public health emergencies rather than the current COVID-19 context.

Reviewer #2: All the comments of the previous review have been addressed adequately by the authors in this submission.

Response: We thank the reviewer for their constructive comments, which have helped us substantially improve the quality and clarity of the manuscript.

---

## [Decision Letter · Decision Letter 2]

16 Feb 2026

“It becomes more difficult when people don’t empathize with us": COVID-19-related stigmatization experienced by survivors in Nepal.

PONE-D-24-58430R2

Dear Dr. Buna Bhandari,

We’re pleased to inform you that your manuscript has been judged scientifically suitable for publication and will be formally accepted for publication once it meets all outstanding technical requirements.

Kind regards,

Habil Otanga, Ph.D

Academic Editor

PLOS One

Additional Editor Comments (optional):

Reviewer comments and suggestions at Round 1 and 2 of revisions are noted as conclusively attempted.

Reviewers' comments:

Reviewer's Responses to Questions

**Comments to the Author**

Reviewer #2: All comments have been addressed

2. Is the manuscript technically sound, and do the data support the conclusions?

Reviewer #2: Yes

3. Has the statistical analysis been performed appropriately and rigorously?

Reviewer #2: Yes

4. Have the authors made all data underlying the findings in their manuscript fully available?

Reviewer #2: No

5. Is the manuscript presented in an intelligible fashion and written in standard English?

Reviewer #2: Yes

Reviewer #2: The comments made by me in the previous version was addressed by the authors and in this version, further valid comments raised by other reviewer has been addressed.

**Do you want your identity to be public for this peer review?** For information about this choice, including consent withdrawal, please see our Privacy Policy

Reviewer #2: No

---

## [Editor Report · Acceptance letter]

PONE-D-24-58430R2

PLOS One

Dear Dr. Bhandari,

I'm pleased to inform you that your manuscript has been deemed suitable for publication in PLOS One. Congratulations! Your manuscript is now being handed over to our production team.

Kind regards,

on behalf of

Dr. Habil Otanga

Academic Editor

PLOS One